# A New Laser-Combined H-Type Device Method for Comprehensive Thermoelectrical Properties Characterization of Two-Dimensional Materials

**DOI:** 10.3390/ma16247680

**Published:** 2023-12-17

**Authors:** Jie Zheng, Shuaiyi Zhao, Haidong Wang, Tianzhuo Zhan

**Affiliations:** 1Department of Engineering Mechanics, Tsinghua University, Beijing 100190, China; zhengj21@mails.tsinghua.edu.cn (J.Z.); zhaosy16@163.com (S.Z.); 2Graduate School of Interdisciplinary New Science, Toyo University, Kawagoe 350-8585, Saitama, Japan; zhan@toyo.jp

**Keywords:** laser heating, H-type device, two-dimensional material, thermoelectric properties

## Abstract

Two-dimensional nanomaterials have obvious advantages in thermoelectric device development. It is rare to use the same experimental system to accurately measure multiple thermoelectrical parameters of the same sample. Therefore, scholars have developed suspended microdevices, T-type and H-type methods to fulfill the abovementioned requirements. These methods usually require a direct-current voltage signal to detect in Seebeck coefficient measurement. However, the thermoelectric potential generated by the finite temperature difference is very weak and can be easily overwritten by the direct-current voltage, thereby affecting the measurement accuracy. In addition, these methods generally require specific electrodes to measure the thermoelectric potential. We propose a measurement method that combines laser heating with an H-type device. By introducing a temperature difference in two-dimensional materials through laser heating, the thermoelectric potential can be accurately measured. This method does not require specific electrodes to simplify the device structure. The thermoelectrical parameters of supported graphene are successfully measured with this method; the results are in good agreement with the literature. The proposed method is unaffected by material size and characteristics. It has potential application value in the characterization of thermoelectric physical properties.

## 1. Introduction

Thermoelectric materials can convert heat and electric energy directly through the thermoelectric effect, which is a pollution-free green-energy mechanism [1,2,3]. In recent years, with the continuous development of new materials and processes, increasingly exceptional thermoelectric materials have been discovered [4,5,6,7]. Among them, two-dimensional materials, such as graphene [8,9,10,11,12] and transition metal dichalcogenides (TMDCs), [13,14,15,16,17] have attracted wide attention in the field of thermoelectrics owing to their extraordinary physical and chemical properties.

Studying the intrinsic thermoelectric properties of materials can provide more detailed experimental basis and theoretical guidance for their application. With most experimental methods, extracting the thermal and electrical parameters of materials simultaneously is difficult; therefore, switching different samples is necessary for achieving multiparameter measurement. Because the sizes of two-dimensional materials are often at the nanometer scale, there are more significant differences between different samples, which poses greater challenges for multiparameter measurement. Therefore, developing an effective and comprehensive measurement method for measuring the thermoelectric properties of two-dimensional materials is of great significance.

The Seebeck coefficient, electrical conductivity, and thermal conductivity are the key parameters in the study of thermoelectric properties. Currently, most commercial multiparameter testing instruments have certain requirements for sample size, such as the thickness requirement to be more than 10 nm. Therefore, specific testing techniques are often used for low-dimensional materials. For example, many scholars have designed some microsized thermoelectric devices for measurement according to the characteristics of micro/nanomaterials [18,19,20,21].

Li et al. [22,23,24,25] measured the thermoelectric parameters of multiwall carbon nanotubes and silicon nanowires by a microfabricated device. The device comprised two silicon nitride (SiNx) films as micropads. A serpentine platinum (Pt) coil was designed on the top of each micropad, which was used as a heater and resistance thermometer to provide Joule heat and calibrate the temperature of the two micropads. The one-dimensional material was bridged between the two micropads. Guang et al. have also successfully measured the thermal conductivity of hybrids of poly(3,4-ethylenedioxythiophene)-tosylate (PEDOT-Tos) and carbon nanotubes (CNTs) by this device [26]. The method could well realize the comprehensive multiparameter measurement of low-dimensional materials. However, the thermal radiation generated by the large micropad area caused a non-negligible effect.

Zhang et al. [27,28,29,30,31] proposed the measurement of the thermal conductivity of single nanowires using the T-type method. The device used in this method has a “T” letter structure. The upper part is a platinum wire bridged between two micropads. The platinum wire acts as a heater and resistance thermometer. The lower part is the third micropad for the heat sink. In the middle part, the nanowire to be tested is bridged between the platinum wire and the third micropad. Joule heat is generated when a constant direct current flows through the platinum wire. The temperature of the sample depends on its thermal conductivity and the heating power of the platinum wire. The governing equation of one-dimensional steady-state heat conduction of the platinum wire and sample was established. The thermal conductivity of a sample could be obtained by solving the equation. However, with this method, measuring the Seebeck coefficient is difficult. Subsequently, Ma et al. [32,33] added a combination of alternating-current heating and direct-current detection based on the original T-type method. They used the modified T-type method to measure the Seebeck coefficient, electrical conductivity, and thermal conductivity of single-crystal Bi_2_S_3_ nanowires.

The H-type method evolved from the T-type method. In contrast to the T-type method, which uses a single metal wire as the sensor, the H-type method employs two metal wires as the sensor. In addition, owing to the limitation of the device’s structure, applying the T-type method to two-dimensional material performance characterization experiments is difficult. By contrast, the H-type method is less restrictive in measuring material structure. This method measures temperature more accurately. Wang et al. [34] measured the thermoelectric properties of a single-crystal cadmium sulfide nanowire using the H-type method. The Seebeck coefficient, thermal conductivity, and electrical conductivity were measured on the same sample under test by simply changing the external circuit. In the same year, they also used this method to measure the thermal conductivity of asymmetrically suspended monolayer graphene [35]. However, the traditional H-type method uses electric heating to measure the Seebeck coefficient, and its heating voltage is on the order of hundreds of millivolts. The thermoelectric voltage is often only a few millivolts. Therefore, the measurement results are easily affected by the heating voltage [34].

In this paper, we propose a measurement method that combines laser heating with an H-type device. Its feasibility has been theoretically proved by numerical simulation [36]. The proposed method is applied for the first time to characterize the thermoelectric properties of single-layer graphene (SLG) supported by a substrate. A temperature difference is introduced between the two ends of the part to be evaluated by laser heating, which solves the problem of the adverse effect of electric heating on the thermoelectric voltage measurement. In addition, laser heating can reduce the additional electrode used to energize the electric heating, thus further simplifying the device structure.

## 2. Experimental Methods

### 2.1. Device Design and Fabrication

Figure 1a,b show the structure diagram and a scanning electron microscopy (SEM) image of the H-type device, respectively. Graphene is placed between two gold (Au) nanowires, labeled A and B in the figures, forming an H-shaped structure. In addition, both ends of the two Au nanowires are connected to Au electrodes with a larger area. Both ends of graphene are also connected to the electrodes. These electrodes are named after the electrode pads below. The parts described above have a silicon substrate underneath, which is used to dissipate heat. In this work, a specific technology was used to fabricate the device [37,38]. SLG was grown on a copper foil by chemical-vapor deposition. The SLG was then transferred to a silicon (Si) wafer with a 100 nm-thick silicon dioxide (SiO_2_) layer. The sample was cut into several 1 × 1 cm chips, and one of these chips was selected for processing in the microelectromechanical system (MEMS) process described below. A 300 nm-thick layer of EB resist (ZEP520A) was spin-coated on the graphene surface. The EB resist was patterned into micrometer-wide strips. The chip was exposed to O_2_ plasma for 2 min, and the graphene not covered by the EB resist was etched away. This layer of EB resist was subsequently removed with butanone. Next, electrodes were fabricated for the chip by the following steps. Another equally thick layer of EB resist was spin-coated on top of the chip and patterned into the shape of the desired electrodes. A 100 nm-thick Au film (on top of a 10 nm-thick chromium adhesive layer) was deposited on the chip using electron-beam physical-vapor deposition (PVD). The EB resist was removed similarly after this step.

After the fabrication of the electrodes, a 600 nm-thick EB resist layer was spin-coated as a protective layer on the SLG. Several windows were created by exposure and development to facilitate the etching of the Si layer on the substrate. Reactive ion etching was employed to etch the SiO_2_ layer not covered by the EB resist. Subsequently, the chip was placed in a XeF_2_ gas reactor to etch the Si substrate and create a 10 μm-deep trench. Finally, the EB resist was removed.

### 2.2. Electrical Conductivity Measurement

To measure electrical conductivity, the circuit shown in Figure 2 was built outside the device. A direct current was passed through the graphene to measure the voltage at both ends and the current flowing through the graphene. The electrical conductivity of graphene, *σ*, can be calculated as
(1)$$\sigma =\frac{IL}{adU},$$
where *U* is the voltage of the graphene, and *I* is the current flowing through the graphene. *L*, *a*, and *d* are the length, width, and thickness of graphene, respectively.

### 2.3. Thermal Conductivity Measurement

Figure 3 shows the circuit and temperature distribution of graphene for measuring thermal conductivity while heating nanowire A. Both nanowires can be used as resistance thermometers. In a certain temperature range, the resistance and temperature of the Au nanowires show a linear trend. Therefore, if the relationship between the resistance and temperature of the Au nanowires is calibrated in advance, the temperature rise of the Au nanowires can be obtained from the change in their resistance. A large current is applied to Au nanowire A, which shows a large temperature rise due to Joule heating. The heating power interval is about 0.1 μW, resulting in a temperature rise of 1 K. The waiting time for heating to record data is about 30 s. At this time, a temperature gradient is generated between Au nanowires A and B. Heat is transferred from A to B through the SiO_2_ support layer and the graphene in the middle. It can be seen from Figure 3b that the total thermal resistance consists of *R*_A_, *R*_s_, and *R*_B_. *R*_A_ and *R*_B_ are the thermal resistance of nanowire A and nanowire B, respectively. *R*_s_ is defined as the thermal resistance composed of the SiO_2_ support layer and the graphene. The thermal conductivity of the SiO_2_ support layer and the graphene determines the amount of heat transferred and the temperature rise of Au nanowire B. Therefore, by measuring the temperature rise of Au nanowires A and B, the effect of SiO_2_ and graphene on thermal conductivity can be obtained. To eliminate the effects of SiO_2_, two steps are required:

Step 1: prepare an H-type device without graphene that contains only SiO_2_. The temperature-rise relationships of Au nanowires A and B are obtained through experimental measurements. Establish a corresponding finite-element model in COMSOL, which contains only SiO_2_ and not graphene. In this model, the only unknown parameter is the thermal conductivity of SiO_2_. By matching the experimental measurement results with the finite element calculation results, the thermal conductivity of SiO_2_ can be obtained.

Step 2: prepare an H-type device containing SiO_2_ and graphene. The temperature-rise relationships of Au nanowires A and B are obtained through experimental measurements. A corresponding finite element model is built in COMSOL, which includes SiO_2_ and graphene. In this model, the thermal conductivity of SiO_2_ is obtained through step 1, and the only unknown parameter is the thermal conductivity of the graphene. By matching the experimental measurements with the finite element calculations, the thermal conductivity of graphene can be obtained.

### 2.4. Seebeck Coefficient Measurement

Figure 4 shows the circuit and temperature distribution of graphene for measuring the Seebeck coefficient during laser heating. The Seebeck coefficient can be calculated by measuring the thermoelectric voltage *V*_tv_ and the temperature difference Δ*T* between the two ends of graphene between Au nanowire A and electrode pad 1. In this work, the electrode pad acted as a heat sink, and the temperature of the low-temperature end connected to the electrode pad could be regarded as being kept at the initial temperature during the measurement process. Therefore, it can be considered that the temperature rise of Au nanowire A (the initial temperature is consistent with that of the electrode pad) before and after heating is the temperature difference Δ*T* between the two ends of the graphene between Au nanowire A and electrode pad 1. The measurement steps are as follows.

Step 1: Measure the resistance of the Au nanowire B by applying a small voltage to it without turning on the laser.

Step 2: Turn on the laser, place the laser at the geometric center of graphene, and deduce measure the resistance of Au nanowire B and *V*_tv_.

Step 3: Calculate the resistance change of Au nanowire B, and further deduce Δ*T*_B_. Because the sample is symmetric and the laser is at the geometric center of the sample, the temperature rise of Au nanowires A and B should be the same, i.e., Δ*T*_A_ = Δ*T*_B_. If the direct measurement of Δ*T*_A_ requires the voltage measurement at both ends of Au nanowire A, the measurement of the thermoelectric voltage will be affected. Therefore, Δ*T*_A_ cannot be measured directly. However, applying voltage to both ends of Au nanowire B does not affect *V*_tv_. Therefore, Δ*T*_A_ is obtained through the measurement of Δ*T*_B_.

Step 4: Establish the finite element model of laser heating in COMSOL, calculate the temperature rise of the high-temperature end of graphene by combining the experimental measurement Δ*T*_A_, and then calculate the Seebeck coefficient S = *V*_tv_/Δ*T*_A_.

## 3. Results and Discussion

Figure 5 illustrates the electrical conductivity of graphene as a function of current at different temperatures. The electrical conductivity of graphene is on the order of 10^5^ Ω^−1^m^−1^. Electrical conductivity increases with increasing current. As the temperature rises, electrons inside the material are excited, which accelerates electron migration. In addition, the scattering of electrons will also increase. Because the electron-migration acceleration effect is greater than the scattering effect in the temperature range of 273 K~373 K, the electrical conductivity increases with a higher temperature when the current is equal.

Figure 6 shows the relationship between Δ*T*_A_ and Δ*T*_B_ in the device containing graphene and the device without graphene and only containing the SiO_2_ support layer when the ambient temperature is 273 K. The fitting reveals that, when the temperature of Au nanowire A rises by 1 K, the temperature rise of Au nanowire B caused by the thermal conduction of SiO_2_ and graphene is 0.0631 K. In the control experiment, the temperature rise of Au nanowire B is 0.0489 K only because of the thermal conduction of SiO_2_. By substituting these two values into the model built in COMSOL, the thermal conductivity was obtained as 645 Wm^−1^K^−1^. The same method can be used to calculate the thermal conductivity of graphene at different ambient temperatures. As shown in Figure 7, our measurement results are similar to those of Seol et al. [39], and the thermal conductivity of graphene shows a decreasing trend with temperature. This trend is explained by the fact that heat transfer in graphene is dominated by phonons. As the temperature increases, the vibration of the graphene lattice is enhanced. This leads to an increase in phonon scattering and a decrease in the average free path of phonons, thereby reducing thermal conductivity.

Figure 8 shows a plot of the change in the Seebeck coefficient of graphene with temperature; the Seebeck coefficient of graphene is ~40 μV/K and exhibits a decreasing trend with an increase in temperature. In the experiment, the Seebeck coefficient is correlated with carrier concentration. When the temperature increases, the electrical conductivity and the carrier concentration in the material increase, resulting in insufficient carrier diffusion. Therefore, the Seebeck coefficient decreases. Li et al. [40] measured the Seebeck coefficient of graphene as a function of the number of layers. They found that the Seebeck coefficient of SLG was ~30 μV/K, which is close in magnitude to the Seebeck coefficient measured in the present work. 

According to the above measurement results, the thermoelectric figure of merit *ZT* of the sample at each temperature can be obtained as follows:(2)$$ZT=\frac{s^{2}\sigma T}{\lambda },$$

At a certain temperature, the electrical conductivity σ will change with the increase of the current. Here, the measured results at each temperature were polynomial fitted. The electrical conductivity at 0.09 mA is substituted into the formula. As shown in Figure 9, the *ZT* value of this sample is on the order of 10^−4^.

## 4. Uncertainty Analysis

### 4.1. Uncertainty in Electrical Conductivity Measurement

The error transfer formula can be obtained from the calculation formula for electrical conductivity:(3)$$\frac{\delta \sigma }{\sigma }=\sqrt{{\left(\frac{\delta U}{U}\right)}^{2}+{\left(\frac{\delta I}{I}\right)}^{2}+{\left(\frac{\delta L}{L}\right)}^{2}+{\left(\frac{\delta d}{d}\right)}^{2}+{\left(\frac{\delta a}{a}\right)}^{2}}.$$

The voltage and current were measured by a high-precision multimeter with an accuracy of eight and a half bits, resulting in an uncertainty of less than 0.01%. The dimensions of graphene were obtained from SEM measurements, with the uncertainty required to be less than 0.1%. In general, the uncertainty in electrical conductivity measurement should be less than 0.2%.

### 4.2. Uncertainty in Thermal Conductivity Measurement

The uncertainty in thermal conductivity measurement mainly includes the following:(1)The uncertainty of the finite element model: nanowires and graphene are important components of the device. In this work, we set the mesh of nanowires and graphene as free triangles with a size of 0.1 μm. To assess the uncertainty of the finite element model in the thermal simulation calculation, grid independence verification was performed to ensure that the influence of the finite element model on the thermal conductivity measurement was less than 0.1%;(2)The uncertainty in the geometric size of graphene and Au nanowires. the geometric dimensions were measured from SEM images with an uncertainty of ~0.1%, which would cause an uncertainty of 1% in the thermal conductivity calculation;(3)The effects of thermal radiation and convection. The entire test process was executed in a vacuum chamber, where the pressure inside the chamber was ~10^−4^ Pa and the thermal convection was negligible. The effects of thermal radiation are described in detail next. The heat loss *J*_r_ by thermal radiation can be estimated by the Stefan–Boltzmann law as follows:
(4)$$J_{r}=\varepsilon_{s}\sigma A_{s}\left(T_{s}^{4}-T_{0}^{4}\right)+\varepsilon_{h}\sigma A_{h}\left(T_{h}^{4}-T_{0}^{4}\right),$$
where *ε*_s_, *A*_s_, and *T*_s_ are the emissivity coefficient, specific surface area, and temperature of graphene; *ε*_h_, *A*_h_, and *T*_h_ are the emissivity coefficient, specific surface area, and temperature of the Au nanowires, respectively; σ is the Stefan–Boltzmann constant; and *T*_0_ is the ambient temperature. In the experiment, the maximum temperature difference between the nanowires used as heaters and the environment was ~50 K. The graphene was approximately 5 μm long and 4 μm wide. Because graphene is a two-dimensional material, the sum of its upper and lower surface areas can be directly considered as its surface area. It follows that $A_{s}=2\times L\times a=4\times 10^{-11}\;m^{2}$, where *L* and *a* are the length and width of graphene, respectively. Nanowires A and B are ~15 μm long, 800 nm wide, and 100 nm thick. Their largest aggregate surface area *A*_h_ can be obtained as $A_{h}=2\times 2\times L_{h}\times \left(a_{h}+d_{h}\right)=5.4\times 10^{-11}m^{2}$, where *L*_h_, *a*_h_, and *d*_h_ are the length, width, and thickness of the nanowires, respectively. The two sides composed of two short sides are six orders of magnitude less than the other four sides; therefore, they are ignored here. If the emissivity coefficient of the nanowires is assumed to be the same as that of graphene (*ε*_h_ = 0.025), the maximum heat loss energy *J*_r_ is calculated to be ~0.001 μW. By contrast, the minimum electrothermal power is ~100 μW, which is five orders of magnitude greater than the radiant heat loss. Therefore, the heat loss from thermal radiation was neglected in this work;

(4)Temperature measurement uncertainty of the H-type sensor as a resistance thermometer. Before the experiment, the H-type sensor was calibrated at different temperatures. The resistance *R* and temperature rise Δ*T* of the sensor follow a linear relationship:
(5)$$\beta_{T}=\frac{\Delta R}{R\Delta T},$$
where *β*_T_ is the resistance-temperature coefficient. According to the error transfer formula, the uncertainty in *β*_T_ can be calculated as follows:
(6)$$\frac{\delta \beta }{\beta }=\sqrt{{\left(\frac{\delta \Delta R}{\Delta R}\right)}^{2}+{\left(\frac{\delta R}{R}\right)}^{2}+{\left(\frac{\delta \Delta T}{\Delta T}\right)}^{2}}=\sqrt{2{\left(\frac{\delta R}{R}\right)}^{2}+{\left(\frac{\delta \Delta T}{\Delta T}\right)}^{2}},$$

Further, the uncertainty in the resistance can be calculated as follows:(7)$$\frac{\delta R}{R}=\sqrt{{\left(\frac{\delta U}{U}\right)}^{2}+{\left(\frac{\delta I}{I}\right)}^{2}}$$
where *U* and *I* represent the voltage and current, respectively, both of which were measured using the Keithley 2002 high-precision digital multimeter with an accuracy of 8.5 bits. The uncertainty in resistance measurement was less than 0.01%. The precision of the temperature control platform was 0.001 K, the maximum temperature rise was controlled to be within 50 K, and the uncertainty of the resistance-temperature coefficient β_T_ was less than 0.1%;

(5)Uncertainty in the measurement of thermal conductivity of Au nanowires and SiO_2_ support layers: the thermal conductivities of the nanowires and SiO_2_ were also considered in the finite element model, and their measurement uncertainties were 1% and 3%, respectively;(6)The influence of thermal contact resistance: the Au nanowires were deposited directly on both sides of the graphene using the electron-beam PVD method without any residue or air in between. Therefore, the thermal contact resistance of our devices was much smaller than that obtained using the ordinary transfer method. The experimental and simulation results showed that the contact thermal resistance of the van der Waals interaction between Au and graphene, *R*_C_, was ~10^−8^ m^2^K/W [41,42,43]. The contact area between the Au nanowires and graphene, *A*_Au-sample_, can be obtained as *A*_Au-sample_ = *a* × *a*_h_ = 3.2 × 10^−12^ m^2^. Therefore, the contact thermal resistance per unit area of the H-type sensor, *R*_Au-sample_, can be calculated as *R*_Au-sample_ = *R*_c_/*A*_Au-sample_ ≈ 3 × 10^3^ K/W.

For comparison, the thermal resistance of graphene along the heat-transfer direction was also calculated. In this study, the measured thermal conductivity (λ) was ~600 W/mK, the heat-transfer length *L*_t_ was 5 μm, and the sample thickness and width were 0.33 nm and 4 μm, respectively. Therefore, the thermal resistance of the sample *R*_s_ could be obtained as *R*_s_ = *L*_t_/*λda* ≈ 6 × 10^6^ K/W, which was three orders of magnitude higher than the thermal resistance of the contact between the nanowires and graphene. Thus, the effect of the contact thermal resistance between the nanowires and graphene was negligible.

After considering various uncertainty sources, the final uncertainty in the measurement of the thermal conductivity of the sample was estimated as ~5%.

### 4.3. Uncertainty in Seebeck Coefficient Measurement

The uncertainty in Seebeck coefficient measurement mainly comes from the uncertainty in thermoelectric voltage measurement with a multimeter and the uncertainty in temperature-difference measurement at both ends of the sample. It has been mentioned previously that the uncertainty in voltage measurement should be 0.01%. The uncertainty in temperature-difference measurement comes from the following aspects.

(1)Temperature measurement uncertainty: the uncertainty of 0.1% can be obtained in the previous statement;(2)The effects of laser positioning: in the measurement process of the Seebeck coefficient, the temperature rise of nanowire A and nanowire B was considered to be approximately the same, i.e., Δ*T*_A_ = Δ*T*_B_. However, the sample center was aligned with a laser by manually controlling it under a microscope. The laser-heating position could deviate from the geometric center of the sample. The accuracy of laser positioning was ~0.1 μm. The distance of laser deviation from the center was set as 0.1 μm in COMSOL, and the temperature rise of Au nanowires A and B was calculated. It was found that Δ*T*_A_ = 21.185 K, Δ*T*_B_ = 19.825 K, and the effect on the measurement was ~6.4%. This step optimization can further improve the laser-positioning accuracy. For example, the stepper motor can be used to control the position of the sample or the laser galvanometer can be used to control the laser heating position;(3)The effect of the thermal conductivity of SiO_2_, Au nanowires, and the sample in the COMSOL simulation: the average temperature rise of nanowire B could be measured experimentally. However, the temperature rise of the high-temperature end of the two-dimensional material needed to be calculated by the finite element method using COMSOL. Accordingly, the thermal conductivities of the Au nanowires, SiO_2_, and graphene were changed in COMSOL to study the influence of the uncertainty in thermal conductivity measurement on the calculation of the temperature rise at the high-temperature end of graphene. The calculation results showed that the effect of the uncertainty in the thermal conductivity measurement on the temperature rise calculation of the high-temperature end of graphene was less than 0.1%;(4)The effect of the actual temperature difference: the Seebeck coefficient varies with temperature. The maximum temperature difference in the experiment is about 20 K. It has little impact on the measurement results. We evaluated it and found that the uncertainty does not exceed 2.8%.

In general, the main factor affecting the accuracy of Seebeck coefficient measurement is the accuracy of laser positioning. The measured total uncertainty in the Seebeck coefficient was ~9.4%.

## 5. Conclusions

To improve the effect of electric heating on the Seebeck coefficient detection results by the traditional H-type method, a comprehensive measurement method for thermoelectric properties combining laser heating with an H-type device is presented herein. The method is not affected by the material characteristics, and the thermoelectric parameters of any suspended or supported two-dimensional material can be accurately measured by the corresponding device-processing method. In addition, the proposed method can reduce the corresponding electrode designed for electric heating in terms of physical structure and further optimize the device structure. The analysis of the experimental results revealed that the uncertainty generated by this method in the measurement of the electrical conductivity, thermal conductivity, and Seebeck coefficient of graphene was within a reasonable range. Among the many factors, the uncertainty caused by the laser-positioning accuracy during the measurement of the Seebeck coefficient has a greater impact on the results. Therefore, improving the laser-positioning accuracy in future studies can further improve the measurement quality of this method.

## Figures and Tables

**Figure 1 materials-16-07680-f001:**
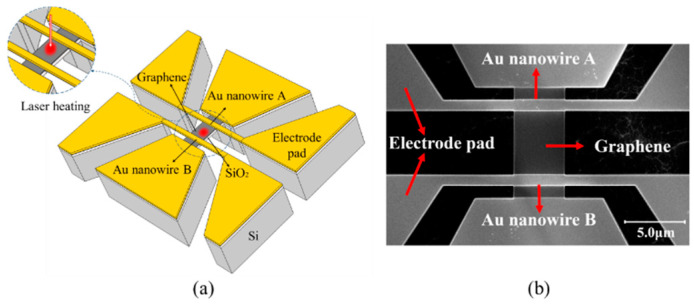
Physical model of the H-type device. (**a**) is the structure diagram of the H-type device, (**b**) is the scanning electron microscopy (SEM) image of the H-type device.

**Figure 2 materials-16-07680-f002:**
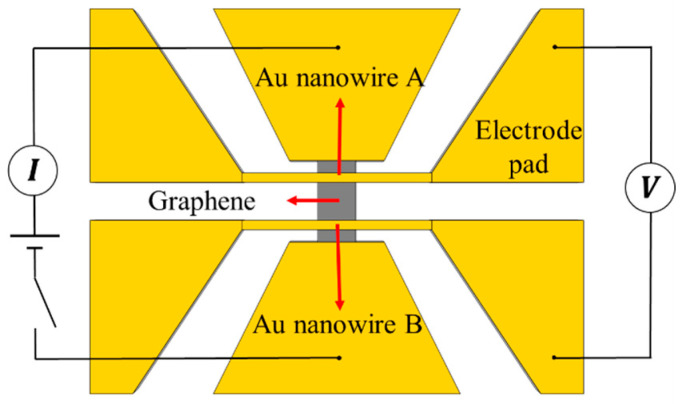
Electrical conductivity measurement circuit diagram.

**Figure 3 materials-16-07680-f003:**
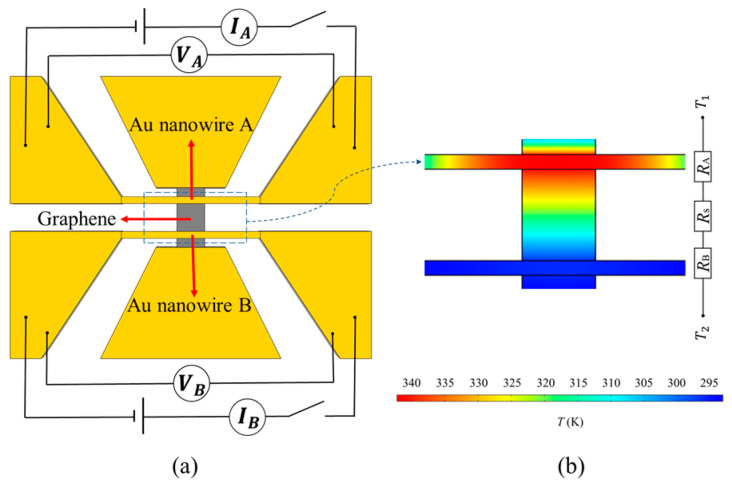
Circuit and temperature distribution of graphene for measuring thermal conductivity while heating nanowire A. (**a**) is the circuit diagram, (**b**) is the temperature distribution diagram.

**Figure 4 materials-16-07680-f004:**
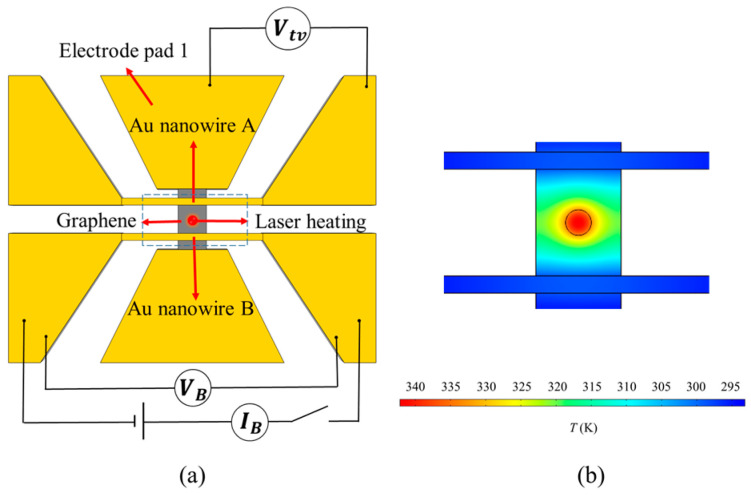
Circuit and temperature distribution of graphene for measuring the Seebeck coefficient during laser heating. The laser was introduced by the Horiba HR Evolution Raman spectrometer with a wavelength of 532 nm and a power of 5 mV. (**a**) is the circuit diagram, (**b**) is the temperature distribution diagram.

**Figure 5 materials-16-07680-f005:**
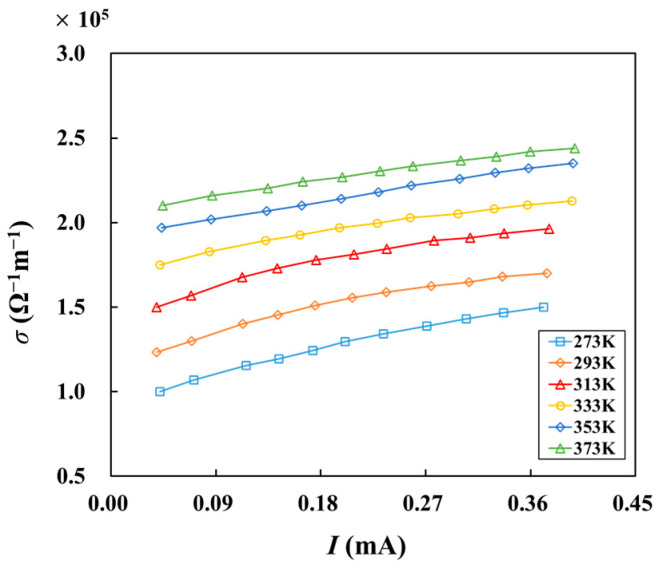
Relationship between electrical conductivity and current at different temperatures.

**Figure 6 materials-16-07680-f006:**
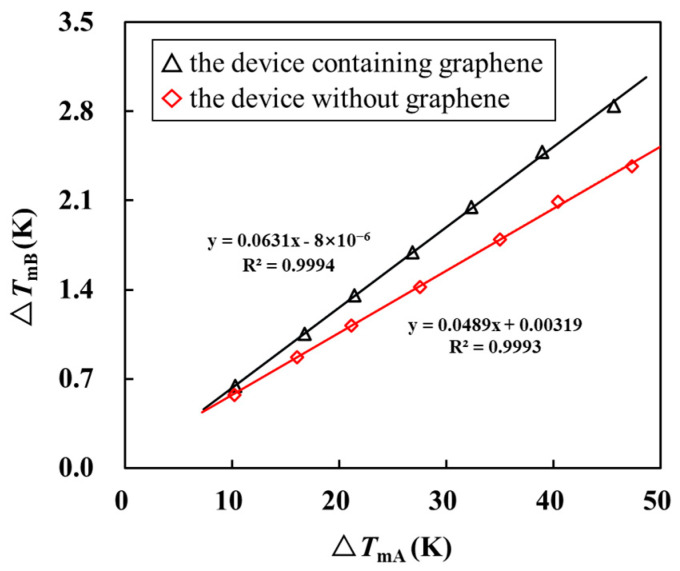
Relationship between Δ*T*_A_ and Δ*T*_B_ in the device containing graphene and the device without graphene and only containing the SiO_2_ support layer when the ambient temperature is 273 K.

**Figure 7 materials-16-07680-f007:**
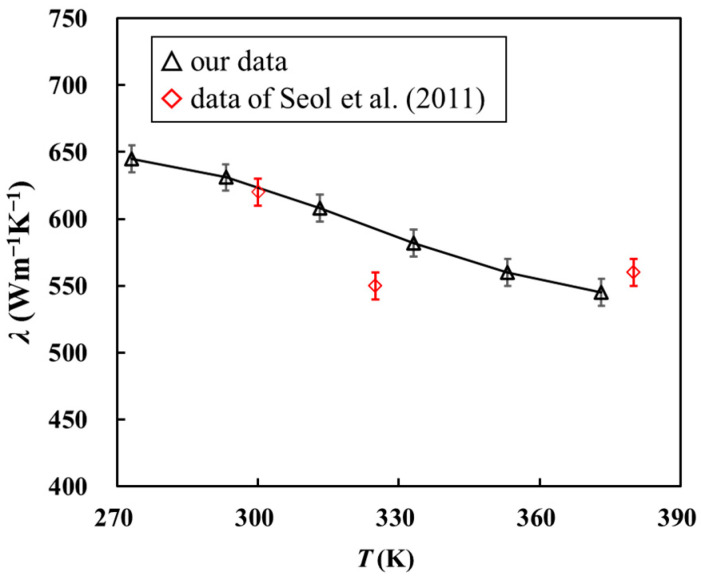
Relationship between thermal conductivity and temperature [39].

**Figure 8 materials-16-07680-f008:**
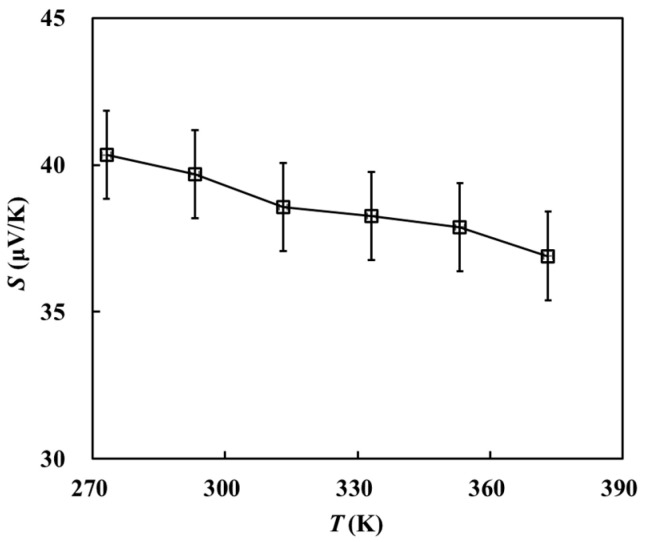
Relationship between the Seebeck coefficient and temperature.

**Figure 9 materials-16-07680-f009:**
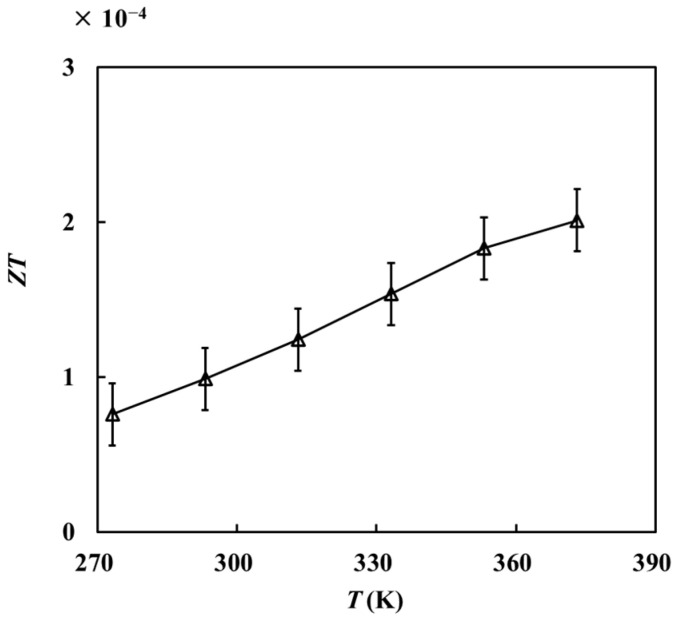
Relationship between the thermoelectric figure of merit and temperature.

## Data Availability

Data are contained within the article.

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
