# Peer review of "A New Laser-Combined H-Type Device Method for Comprehensive Thermoelectrical Properties Characterization of Two-Dimensional Materials"

_materials, 2023, doi:10.3390/ma16247680_

Round 1

Reviewer 1 Report

Comments and Suggestions for Authors

Please see the attached comment file.

Author Response

Dear reviewer,

Thank you very much for your valuable comments. Attachment is our response.

Sincerely yours,

Jie Zheng

Reviewer 2 Report

Comments and Suggestions for Authors

The objective of the work consists in the development of a new method based on laser-combined H-type device for obtaining a comprehensive characterization of the thermoelectrical properties of two-dimensional materials, in the specific reported case graphene.

As stated by the authors, this work could lay the groundwork for the improvement in the characterization of physical properties of thermoelectric materials, which represent a pollution-free green energy mechanism. It is also worth noting that the proposed method is unaffected by material size and characteristics, as discussed in the paper.

In my opinion, the quality of the manuscript is high in terms of perspectives, characterizations performed , process explanations and readibility.

The only major issue regards Fig. 6: in my opinion the measure units presented in the graph axis are wrong, not in agreement with the values previously reported in the paper (lines 208-211). 

Here a couple of further comments:

1) It could be interesting giving further details regarding COMSOL simulations (grid, values parameters employed)

2) Figure 5. Relationship between electrical conductivity and current at different temperatures. Is there any explanation for this trend? If yes, it would be interesting to addi t in the paper. Could it be interesting to fit the data with a curve?

3) Decreasing trends are also observable in Fig. 7 and Fig. 8: would it be possible to give an explanation to this behaviour of thermal conductivity and Seeback coefficient?

Author Response

(The authors gave the same response as above.)

Reviewer 3 Report

Comments and Suggestions for Authors

The authors report a new method for an accurate measurement of thermoelectric properties, using an H-type device and laser heating. The experimental device tested in this study contains a graphene monolayer transferred on a SiO2 substrate. The details of the device fabrication process and of the measurement of the electrical and thermal conductivities are indicated. For the latter, a two-step procedure with COMSOL modeling is proposed, in order to extract the contribution of the graphene alone. Then the measurement procedure of the temperature difference (\Delta T_A) and of calculation the Seebeck coefficient is discussed. Finally, the uncertainties of measuring both electrical and thermal conductivities are are analyzed.

The paper presents an interesting approach concerning the improvement of the thermolectric measurements. It is clearly written and the calibration results are compatible with the existing literature data (e.g. thermal conductivity of graphene). However there some issues that need to be addressed:

- The temperature difference \Delta T_A = 21.185 K seems rather large. Usually the Seebeck coefficient is defined taking into account small temperature differences (resulting in small thermoelectric voltages, S = dV_T / dT). In thermoelectric generators the temperature difference can be larger to yield a large thermoelectric voltage. However, here, the authors pursue the measurement of the Seebeck coefficient.

- It is typically necessary to have a stable thermal map (gradients). The authors should describe in more detail how the global temperature change was performed (e.g. waiting times between consecutive measurements) and the validity of the results with respect to equilibration.

- In the conclusions it is mentioned that "The method is not affected by the material characteristics, and the thermoelectric parameters of any _suspended_ or supported two-dimensional material can be accurately measured"
Can the authors unambiguously claim that measurements with suspended 2D materials would be similarly accurate ?

- The thermoelectric figure of merit (ZT) of the device can be added in the results section.

Author Response

(The authors gave the same response as above.)

Round 2

Reviewer 3 Report

Comments and Suggestions for Authors

The authors included the suggestions of the refeere and the revised manuscript has improved.

There are still two brief comments:

Line 257: "The electrical conductivity at 0 mA is substituted into the formula."
--> in the zero current limit, at the open circuit limit

Line 372: "It has _a little_ impact on the measurement results."

Author Response

(The authors gave the same response as above.)
